# Selectivity of Current Extraction Techniques for Flavonoids from Plant Materials

**Milena Tzanova** [1,*] , **Vasil Atanasov** [1] , **Zvezdelina Yaneva** [2] , **Donika Ivanova** [2]
**and Toncho Dinev** [1]

1   Department of Biochemistry, Microbiology and Physics, Faculty of Agriculture, Trakia University,
    6000 Stara Zagora, Bulgaria; vka@mail.bg (V.A.); dinev_sz@mail.bg (T.D.)
2   Department of Pharmacology, Animal Physiology and Physiological Chemistry,
    Faculty of Veterinary Medicine, Trakia University, 6000 Stara Zagora, Bulgaria; z.yaneva@abv.bg (Z.Y.);
    donika_georgiewa@abv.bg (D.I.)
*   Correspondence: tzanova_m@yahoo.com; Tel.: +359-42-699-315

**Abstract:** Flavonoids have a broad spectrum of established positive effects on human and animal health. They find an application in medicine for disease therapy and chemoprevention, whence the interest in flavonoids increases. In addition, they are used in food and cosmetic industries as pigments and biopreservatives. Plants are an inexhaustible source of flavonoids. The most important step of plant raw material processing is extraction and isolation of target compounds. The quality of an extract and efficiency of a procedure are influenced by several factors: Plant material and pre-extracting sample preparation, type of solvent, extraction technique, physicochemical conditions, etc. The present overview discusses the common problems and key challenges of the extraction procedures and the different mechanisms for selective extraction of flavonoids from different plant sources. In summary, there is no universal extraction method and each optimized procedure is individual for the respective plants. For an extraction technique to be selective, it must combine an optimal solvent or mixture of solvents with an appropriate technique. Last but not least, its optimization is important for a variety of applications. Moreover, when the selected method needs to be standardized, it must achieve acceptable degree of repeatability and reproducibility.

**Keywords:** flavonoids; extraction technique; selective; plant

## 1. Introduction

Flavonoids are small molecules, produced de novo by plants as secondary metabolites in response to diverse biotic and abiotic factors. These chemical compounds have a broad spectrum of established health-promoting effects [1–4]. They are due to their antioxidative, anti-inflammatory, anti-mutagenic, and anti-carcinogenic properties coupled with their capacity to modulate key cellular enzyme functions. They are also known as potent inhibitors of several enzymes, such as xanthine oxidase (XO), cyclo-oxygenase (COX), lipoxygenase, and phosphoinositide 3-kinase [5,6]. The antioxidant potential of flavonoids is due to their ability to scavenge free radicals, as indicated by Korkina and Afanasev [7]. Kerry and Abbey [8] reported that by scavenging radicals, flavonoids can inhibit low-density lipoprotein (LDL) oxidation in in vitro studies. They further mentioned that this action protects the LDL particles and, theoretically, flavonoids may have preventive action against atherosclerosis. Flavonoids can build complex by chelation metal ions, which can be crucial in the prevention of radical generation, which damages some target biomolecules [2,9]. In a review, Cushnie and Lamb (2006) described their antibacterial, antifungal, and antiviral activity, as well as the synergy between the antimicrobial activity of several groups of flavonoids and between flavonoids and existing chemotherapeutics [10].

Because of their positive effects on human and animal health and medical application for disease therapy and chemoprevention, interest in flavonoids is increasing [1,3,4]. Furthermore, they are used in the food and cosmetic industries as pigments and biopreservatives [8].

Flavonoids are widely distributed chemical compounds in the plant kingdom. Plants are therefore an inexhaustible source of flavonoids. The most important step in plant raw materials processing is the extraction and isolation of target compounds. The quality of an extract and the efficiency of a procedure are influenced by several factors: Plant material and pre-extracting sample preparation, solvent type, extraction technique, physicochemical conditions, etc. The classical methods (maceration and Soxhlet extraction) are generally used in research laboratories or in small manufacturing companies. Microwave-assisted extraction (MAE), ultrasound-assisted extraction (UAE), and supercritical fluid extraction (SFE) belong to a group of modern methods. They are very promising: The yield is increased at a lower cost [11].

A lot of research teams compared classical modern methods for the extraction of flavonoids from plant materials and discussed their advantages and disadvantages [4,11–16]. But the focus was not on the selectivity of the reviewed methods. Moreso, the plant matrix is complex, and the extraction produces a cocktail of substances. When the goal is to produce an extract with a narrow range of physiological effects, the first step of sample preparation and extraction is critical. Therefore, the solvents and the technique for extracting must be carefully selected. The potential role of selective extraction methods has a significant importance for the subsequent steps of the extract production [17]. The selectivity of this first extraction step contributes to the effectivity, profitability, and simplicity of the whole process of extract producing. This overview discusses common problems and key challenges in extraction procedures, as well as various mechanisms for selective extraction of flavonoids from different plant sources.

## 2. Flavonoids—Classification and Natural Sources

The flavonoid family includes more than 6000 low-molecular-weight phenolic compounds [18], derivatives of flavan (Figure 1). The main subgroups are flavones, flavonols, flavonones, flavononols, flavan-3-ols, anthocyanines, isoflavones, and chalcones.

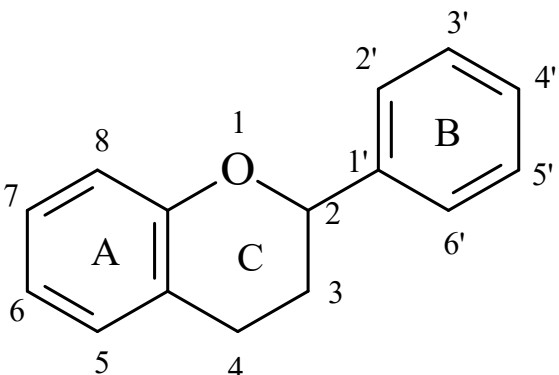

**Figure 1.** Flavan—basic structure of flavonoids.

Their biochemical roles are multifarious: From flower pigmentation to taking part in the growing processes of the plant organism, and the defense against diseases [19]. Each flavonoid group plays a unique biochemical role and has a particular distribution in plants [20]: Anthocyanins are responsible for the coloring of the plants, and are therefore found in high amounts in the plant flowers and the skins of colored fruits; flavan-3-ols, such as catechin, epicatechin gallate, etc., are found in high concentrations in green tea; isoflavones are found in legumes (e.g., soybean); flavanones, in citrus fruits; flavones, in green leafy spices (e.g., parsley); and flavonols, in most plants. Chalcones are a subclass of flavonoids. They are characterized by the absence of "ring C" of the basic skeleton structure. Chalcones can be referred to as open-chain flavonoids. Major examples of chalcones include phloridzin,

arbutin, phloretin, and chalconaringenin. Chalcones occur in significant amounts in tomatoes, pears, strawberries, bearberries, and certain wheat products [3]. The most popular edible and medicinal plants rich in flavonoids, categorized in subgroups, are systematized in Tables 1 and 2.

**Table 1.** Food plants rich in flavonoids.

| Flavonoid Group & Basic Structure | Flavonoids | Food | References |
|---|---|---|---|
| | Kaempferol; quercetin; myrecitin; (−)-epicatechin | Black berries; wine | [21] [22] [23] |
| | Kaempferol; quercetin | Tomato | [24] [25] |
| | (+)-Catechin; (−)-epicatechin; epigallocatechin; chrysin; apigenin; quercetin; kaempferol | Tea | [26] [27] [28] |
| Flavonols and Flavan-3-ols | (+)-Catechin; (−)-epicatechin; quercetin | Coffee; cocoa; apple | [22] [29] [30] [31] [32] [33] |
| | Kaempferol; quercetin; myricetin; tamarixetin | Onion; red wine; olive oil; berries; grapefruit; orange | [34] [35] |
| | (+)-Catechin; (−)-epicatechin; quercetin; kaempferol | Red berries; strawberries | [22] [36] [37] [38] |
| | Quercetin | Lemon; olive; aspargus | [35] [39] [40] |
| | Kaempferol | Saffron spice | [41] |
| | Kaempferol; quercetin | Broccoli; brussel sprouts | [28] [35] |
| | (+)-Catechin; (−)-epicatechin | Apricot; nectarine; peach; plum; fig; banana; kiwi; hazelnut | [23] [22] [42] [43] [44] |
| | (+)-Catechin; (−)-epicatechin; quercetin; isorhamnetin; kaempferol | Almond | [44] [45] |
| Flavones | Luteolin | Fruit skins; red wine; buckwheat; red pepper; tomato skin; lemon; watermelon; brussel sprouts; pumpkin | [46] [47] [34] [35] |
| | Luteolin; apigenin; isorhoifolin | Olive | [39] [48] |

**Table 1.** *Cont.*

| Flavonoid Group & Basic Structure | Flavonoids | Food | References |
|---|---|---|---|
|  Flavonones | Naringin; eriodictyol | Almond | [45] |
| | Naringin; maringenin; taxifolin; hesperitin; eriodictyol | Citrus fruits; grapefruit; lemon; orange | [49] [50] [51] [35] |
|  Isoflavones | Daidzin; genistein; glycitin; sissotrin; ononin | Soya bean | [52] [53] [54] [55] |
| | Genistin; daidzin; biochanin A | Peanut | [56] |
| | Biochanin A; formononetin | Red clover | [55] |
|  Anthocyanins | Apigenidin; cyanidin | Cherry; easberry; strawberry | [34] [47] [38] |
| | Cyanidin | Olive | [39] [48] |

**Table 2.** Medicinal plants rich in flavonoids.

| Flavonoid Group & Basic Structure | Flavonoids | Medicinal Plant (Family) | References |
|---|---|---|---|
|  Flavonols and  Flavan-3-ols | (+)-Catechin | *Brysonima crassa* (Compositae) | [57] |
| | Isorhamnetin | *Calendula officinalis* (Compositae) | [58] |
| | Kaempferol | *Acalypha indica* (Euphorbiaceae); *Clitoria ternatea* (Fabaceae); *Pteris vittata L* (Pteridaceae) | [59] [60] [61] |
| | Quercetin | *Betula pendula* (Betulaceae) *Bauhinia monandra* (Fabaceae); *Pteris vittata L* (Pteridaceae) *Cannabis sativa* (Compositae); *Azadirachta indica* (Meliaceae); *Angelica* L. (Apiaceae); | [58] [61] [62] [63] [64] |

**Table 2.** *Cont.*

| Flavonoid Group & Basic Structure | Flavonoids | Medicinal Plant (Family) | References |
|---|---|---|---|
| | Hyperoside | *Tilia cordata* (Tiliaceae) | [58] |
| | Isoquercetin | *Mimosa pudica* (Mimosoideae) | [65] |
| | Pongaflavonol | *Pongamia pinnata* (Fabaceae) | [66] |
| | 2-(3, 4-dihydroxyphenyl)-3,5,7-trihydroxy-4H-chromen-4-one | *Chenopodium album* L. (Chenopodiaceae) | [67] |
| | 2-(3,4-dihydroxy-5-methoxy-phenyl)-3,5-dihydroxy-6,7-dimethoxychromen-4-one | *Euphorbia neriifolia* (Euphorbiaceae) | [68] |
| Flavones | Pectolinarigenin | *Clerodendrum phlomidis* (Verbenaceae) | [62] |
| | Luteolin | *Aloe vera* (Asphodelaceae); *Momordica charantia* (Curcurbitaceae); *Bacopa moneirra* (Scrophulariaceae); *Angelica* L. (Apiaceae); *Mentha longifolia* (Lamiaceae) | [59] [63] [69] [70] |
| | Hispidulin; apigenin; cirsimaritin | *Rosmarinus officinalis* L. (Lamiaceae) | [71] [72] |
| | Luteolin; hispidulin; apigenin; cirsimaritin | *Salvia officinalis* L. (Lamiaceae) | [71] |
| | Luteolin; hispidulin | *Thymus* L. (Lamiaceae) | [71] [73] |
| | Apigenin; hispidulin | *Verbena officinalis* L. (Verbenaceae) | [73] |
| | 5-hydroxy-7,8-dimethoxyflavone | *Andrographis paniculata* (Acanthaceae) | [58] |
| | 3,4-methlenedioxyflavone | *Limnophila indica* (Scrophulariaceae) | [65] |
| | Chrysin | *Oroxylum indicum* (Bignoniaceaea) | [65] |
| | Vitexin | *Passiflora incarnate* (Passifloraceae) | [58] |
| Flavonones | Narginin | *Rosmarinus officinalis* L. (Lamiaceae) | [71] [72] |
| | Hesperidin | *Citrus medica* (Rutaceae) | [59] |
| | Liquiritin | *Glyccheriza glabra* (Leguminosae) | [58] |
| | Kurarinol; kurarinone | *Sophora flavescens* Ait. (Fabaceae) | [74] |

**Table 2.** *Cont.*

| Flavonoid Group & Basic Structure | Flavonoids | Medicinal Plant (Family) | References |
|---|---|---|---|
| Flavononols | Kushenol I; kushenol N | *Sophora flavescens* Ait. (Fabaceae) | [74] |
| Isoflavones | Genistein | *Calopogonium muconoides* (Fabaceae) *Butea monospermea* (Fabaceae); *Andira macrothyrsa* (Fabaceae); | [55] [64] |
| | Biochanin A | *Cratylia argentea* (Fabaceae); *A. macrothyrsa* (Melastomataceae) | [55] |

## 3. Sample Preparation

Before the beginning of the extraction procedure, the following issues have to be clarified: First—the plant material type, and second—the potential application of the flavonoid extract (Figure 2). In case the plant material is used to produce a standard extract—an extract with established biochemical activity and pharmaceutical application—the standard extraction procedure has to be followed, in order to imitate the traditional "herbal" drug [75]. When the plant is not well known, or the aim of the procedure is the selective separation of the flavonoid fraction—a fraction free of fats, terpenoids, pigments, saponines, alcaloids, etc.—the pre-existing extraction procedure has to be modulated into new one. In this respect, scientific knowledge in this area can be very helpful.

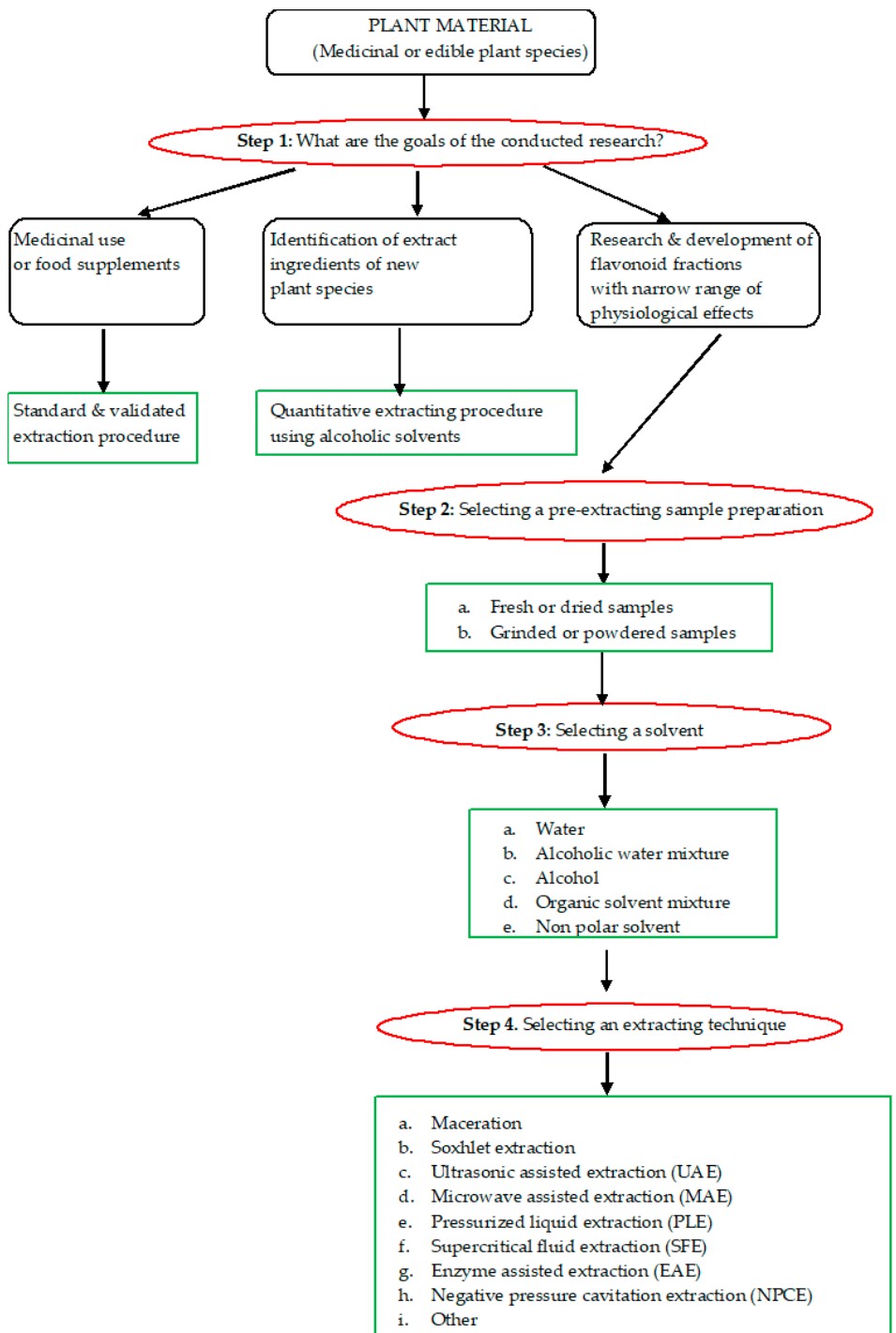

**Figure 2.** Flavonoid extraction scheme.

Sample preparation is the first step in the extraction procedure and usually includes two or sometimes three steps: Drying, grinding, and defatting.

Flavonoids, particularly glycosides, can be degraded by enzymes when the plant material is fresh and undried [76]. Later on, the first step is to dry, lyophilize, or freeze the sample. Often the yield of flavonoid conjugates is reduced by the usage of dried plant material. For example, acylated flavonoid glycosides are particularly thermally unstable and are degraded during the process of drying at high temperature, and this first step of the sample utilization is very important for the profiling of this class of natural products in studies of their physiological and biochemical roles in plants [12]. Plant samples,

such as leaves, barks, roots, fruits, and flowers, can be extracted from fresh or dried plant material. Often a dried sample is preferred because of the experimental design timing [77].

There are several methods of drying the plant material—air-, microwave-, oven-, and freeze-drying (lyophilisation). Because of their effectivity and efficiency air- and freeze-drying are preferred. Using lyophilization, the phytochemicals are preserved, but this technique is complex and expensive [11]. Air-drying usually takes a long time but does not need any lab equipment, and the heat-labile compounds are preserved. For that reason, shade-drying was preferred by many researchers [68,78,79]. Microwave- and oven-drying require a short time but sometimes cause degradation of plant products and can be applied for drying of no conjugated and thermo-stable flavonoids. Oven-drying at 40–45 °C is considered as a compromising plant material preparation. It is easy to be carried out, and is time consuming [80]. However, even a short time of exposure to high temperatures can destroy some thermo-unstable compounds, and the extract profile is changed [57].

After drying, the plant material must be ground, whereby the cells are destroyed, and the ingredients are able to leach out. Decreasing the particle size enhances surface contact between samples and extraction solvents. A particle size smaller than 0.5 mm is optimal for efficient extraction [81]. The usage of powdered dried plant material leads to effective extraction of flavonoids [67,74,79,82,83].

Often, the target compounds of extracting are unconjugated flavonoids, which are the active compounds [84]. In this case, fresh plant material can be used, which is directly ground and suspended in the selected solvent or mixture of solvents. Flavonoids are extracted without drying and grinding from fruits, such as citrus [85], or from plants, whose pre-extracting preparation includes enzymatic treatment [86].

The presence of lipophilic substances can affect the profile of the composition of flavonoids and their derivatives in the extracts obtained, and in many cases an additional cleaning procedure (e.g., solid-phase extraction) is required [12]. Many researchers started with removing lipophilic compounds from the plant material by using n-hexane [87–89] or petroleum ether [67]. The non-polar flavonoids are passed into the liquid fraction. Next follows the extraction with a polar, alcoholic solvent to mobilize the polar flavonoid fractions. The extraction of flavonones and anthocyanes depends on the pH values [76].

## 4. Extraction Techniques

The choice of the extraction procedure for flavonoids from plant material is crucial and depends on the goals of the conducted research [14]. Some questions have to be answered: Is this fraction polar or non-polar? Are the flavonoids thermally stable or unstable? What amount of extract should be made? What equipment is available and accessible?

Extraction of flavonoids is designed using the similar principles of polyphenols extraction. The process is usually carried out with methanol, ethanol, acetonitrile, acetone, or their mixtures with water [90]. The required solvent polarity depends on the type of the flavonoid [12]. For less polar flavonoid structures such as isoflavons, flavonones, and flavones, the right choice can be acetone, chloroform, methylene chloride, and diethyl ether; for more polar flavonoid fractions, the solvent is usually alcohol or an alcohol–water mixture [76].

The last step is the selection of the extraction technique. There are several options divided into two groups: Conventional techniques (maceration, reflux, and Soxhlet extraction) and modern techniques (extraction by ultrasound, microwaves, pressurized liquid, supercritical fluid, enzyme assistance, matrix solid-phase dispersion, etc).

### 4.1. Conventional Extraction Techniques

The conventional methods are easy to perform, need no special equipment, are applicable for extraction of a large number of samples, and lead to high yield of the extract obtained. However, they are not selective. Some selectivity can be reached with the right solvent, according to the polarity of the target flavonoid fraction.

#### 4.1.1. Maceration

Maceration is carried out by soaking the ground plant materials in a stoppered container with a proper solvent. The sample is allowed to stand for at least 3 days at room temperature and is shaken frequently. During the process, the solvent softens and breaks the plant's cell wall and the soluble phytochemicals are released. Afterwards, a filtration of the mixture follows. Table 3 presents a number of studies in which flavonoid fractions were successfully extracted by conventional techniques, including maceration.

**Table 3.** Conventional techniques used in the recent years for extraction of flavonoids.

| Flavonoids | Source | Solvent | Reference |
|---|---|---|---|
| Extraction technique: Maceration | | | |
| Flavonol | *P. vittata* L. | Methanol | [61] |
| Total flavonoids | *Moringa oliefera* | 70% ethanol in water | [77] |
| Total flavonoids | *Acanthospermum hispidium* | Methanol | [79] |
| Total flavonoids | *Ganoderma lucidum* | 75% acetone in water | [83] |
| Flavones and flavonones | *Citrus fruits* | Water + HCl; methanol; ethanol; acetone; ethyl acetate; n-hexane | [85] |
| Total flavonoid | *Mitracarpus hirtus* | Methanol | [89] |
| Total flavonoid | *Dendrobium officinale* | 78% ethanol in water | [91] |
| Total flavonoids | *Cosmos caudatus* | 70% ethanol in water | [92] |
| Total flavonoids | Cassia angustifolia | 70% ethanol in water | [93] |
| Isoflavones | Chickpea | 80% methanol + HCl | [94] |
| Isoflavones | *Ptycholobium contortum* | CH2Cl2 + methanol | [95] |
| Isoflavones | *Dalbergia odorifera* T. Chen | 80% ethanol in water | [96] |
| Isoflavones | *Amorpha fruticosa* L. | Acetone | [97] |
| Isoflavones | *Pueraria lobata* (Willd.) Ohwi | Butanol + water | [98] |
| Isoflavones | *Pueraria lobata* | 70% etahanol | [99] |
| Isoflavones | *P. lobata* | Water | [100] |
| Extraction technique: Reflux | | | |
| Isoflavones | Quinoa seeds | Methanol + 0.1% butylated hydroxytoluene + hydrochloric acid (4:1) | [101] |
| Isoflavones | *Pueraria lobata* (Willd.) Ohwi root | Methanol | [102] |
| Extraction technique: Soxhlet extraction | | | |
| Flavonols | *Chenopodium album* L. | Acetone | [67] |
| Flavonols | *Euphorbia neriifolia* | 70% ethanol in water | [68] |
| Total flavonoids | *Moringa oliefera* | 70% ethanol in water | [77] |
| Flavones | *Orthosiphon stamineus* | Methanol | [82] |
| Total flavonoids | *Cassia angustifolia* | 70% ethanol in water | [93] |
| Flavonols and flavones | *Mentha spicata* L. leaves | 70% ethanol in water | [103] |
| Isoflavones | *D. oliveri* | Hexane; dichloromethane; ethyl acetate; and methanol | [104] |

In the conventional methods, the choice of the solvents determines the type of the target compounds and the extraction rate [14]. Cowan [105] pointed out acetone as the most selective solvent for extracting flavonoids. Often alcohol–water mixtures are applied for the extraction of flavonoids and their conjugates from plant material. Seventy percent methanol is efficient solvent for extracting flavonoids by maceration [77,92,93,99]. Maceration is the preferred method for extraction of citrus flavonoids: Raw material can be used either fresh or dry; solvents, such as acidified water, methanol, ethanol, acetone, ethyl acetate, and n-hexane are applied for the extraction, of which methanol is frequently used [85].

The selection of suitable operational conditions for a new developed method of extraction is of great importance. For example, the intensity of temperature and light must be evaluated for extraction of thermo-labile compounds. Another risk factor is pH value. Vankar and Srivastava used a slightly acidic solvent (0.1% HCl in methanol *v/v*) to extract anthocyanin from different colored flowers and described the effect of pH on the extraction procedure [106]. Compared to the acetic acid, the hydrochloric acid in ethanol was found to be more efficient for the extraction of anthocyanin [107]. Similar results were obtained using 70% ethanol during total flavonoids extraction from *Moringa oliefera* [77]. The most influential extraction factors in the newly developed methods are the solvent type. However, it is reported that the variation of the solvent–sample ratio does not significantly change the extraction effect and the superfluous solvent amount can be saved. Each optimized method is individual for the respective plants. All factors (temperature, solvent type, duration, etc.) can have a negative impact on the extraction procedure, and the lack of proper parameter optimization may cause degradation of the phytochemicals.

The extracts obtained by maceration are complex mixtures and it is necessary to carry out a further purification by re-extrication [61,79,94], column chromatography [85,89,100], or a combination of both methods [95,96].

The technique of maceration is the easiest and no special lab equipment is required. The main disadvantages of this method are the large volume of solvents, the long processing time, and the need for future purification. In case purity is an issue, advanced extraction technology should be considered.

### 4.1.2. Reflux and Soxhlet Extraction

Reflux and Soxhlet extraction are high-temperature continuous extraction procedures. Soxhlet extraction is carried out using Soxhlet apparatus: A ground sample is placed in a porous "thimble", which is located in a chamber. Extraction solvents are heated in the bottom flask, vaporized into the sample thimble, condensed in the condenser, and dripped back. Compared to maceration, these methods require a smaller volume of solvent [11] and shorter processing time. The following general disadvantages of Soxhlet extraction are: The extracted flavonoids must be thermostable, the plant sample should be dried, and toxic and flammable liquid organic solvents are used, which is a potential risk factor. A few factors (temperature, solvent–sample ratio, and agitation speed) must be evaluated [108].

A few examples of successful flavonoid extraction from plant material by reflux and Soxhlet methods are given in Table 3. A solvent mixture of methanol, 0.1% butylated hydroxytoluene, and hydrochloric acid (4:1) was used for quantitative extraction of two major isoflavones from quinoa seeds by reflux [101]. Mun and Mun [102] extracted isoflavonoids from *Pueraria lobata* (Willd.) by reflux in methanol. The Soxhlet extraction of *Moringa oliefera* leaves led to a lower yield of phenolic amount, including flavonoids [77]. Pluempanupat et al. [104] extracted isoflavonoids from air-dried and powdered *Dalbergia oliveri* heartwood by Soxhlet. The researcher used solvents with different polarity (hexane, dichloromethane, ethyl acetate, and methanol). The extraction started with the low polar solvent and finished with the high polar solvent. Thus, selectivity of some thermally stable flavonoids can be attained by Soxhlet extraction. However, different solvents have to be used for several successive extractions. Powdered *Clitorea ternate* flowers were defatted by Soxhlet using petroleum ether at 60–80 °C. After further extraction with ethanol, alkaloids, and saponins were determined,

but the major component, anthocyanin, was absent. The researchers explained this with its oxidation and degradation during the extraction process [109].

The extracts obtained by Soxhlet extraction have a complex composition and the flavonoid fraction must be re-extracted to be purified [82] or could be purified by liquid chromatography [67,68]. Compared to soaking, the advantages of Soxhlet extraction are the shorter processing time, and the possibility for automation. Nevertheless, the degradation of thermally instable compounds is a serious problem.

After quantitative extraction by classical solid–liquid methods like Soxhlet or maceration, the purification in one step can lead to the preparation of extract rich in flavonoids. Such selective options use nano-encapsulated processing [110], as well as polymer-based solid-phase extraction [16]. The sorbents should have high selectivity according to the flavonoid substances or membrane extraction.

## 4.2. Green Extraction Techniques

The major shortcomings of the conventional extraction techniques are a long extraction time, a requirement of costly solvent, evaporation of a huge amount of solvent, low extraction selectivity, and these techniques are difficult to be automated [14]. New and promising techniques are being developed to break through the limitations of classical extraction methods. They are categorized as non-conventional, "green" extraction techniques. They have a lot of advantages: Reduction of organic solvent consumption and sample degradation, pollution prevention, elimination of additional sample clean-up and concentration steps, improvement of extraction efficiency and selectivity, and ability for automation [111].

Some of the most promising techniques are ultrasound-assisted extraction (UAE), enzyme-assisted extraction (EAE), microwave-assisted extraction (MAE), pressurized liquid extraction (PLE), supercritical fluid extraction (SFE), enzyme-assisted extraction (EAE), and matrix solid-phase dispersion (MSPD). There are individual solution cases of flavonoid extraction that demonstrated a promising selectivity. Green extraction techniques concerning the isolation of flavonoids from plant matrices are listed in Table 4.

**Table 4.** Green techniques used in the recent years for extraction of flavonoids.

| Flavonoids | Source | Solvent | Extraction Technique | Reference |
|---|---|---|---|---|
| Anthocyanins | *Mirabilis jalab* L. | Methanol/ethanol + 0.1% HCl aq | UAE | [112] |
| Total flavonoids | *Cryptotaenia japonica* Hassk | Ethanol | UAE | [113] |
| Isoflavones | *Pueraria lobata* (Wild.) Ohwi | 50% ethanol in water | UAE | [114] |
| Isoflavones | *Pueraria lobata* (Wild.) Ohwi | 40% ethanol in water | UAE | [115] |
| Isoflavones | *P. lobatae*; *P. thomsonii* | Ethanol + water | UAE | [116] |
| Isoflavones | *Iris tectorum* | 70% methanol in water | UAE | [117] |
| Isoflavones | *Radix astragali* | Methanol | UAE | [118] |
| Total flavonoids | Citrus peels | Methanol | UAE | [119] |
| Total flavonoids | Grapes | 50% methanol in water | UAE | [120] |
| Isothiocyanates and flavanols | Chilean papaya | 80% methanol in water | UAE | [121] |
| Total flavonoids | Orange peels | Ethanol:water (4:1) | UAE | [122] |
| Anthocyanins | Jabuticaba | Ethanol | UAE | [123] |
| Total flavonoids | Olive leaves | 50% ethanol in water | UAE | [124] |

**Table 4.** *Cont.*

| Flavonoids | Source | Solvent | Extraction Technique | Reference |
|---|---|---|---|---|
| Total flavonoids | Wheat | 65% methanol in water | UAE | [125] |
| Isoflavones | Soya bean | Water + acetonitrile + HCl | High-power UAE | [126] |
| Total flavonoids | *Radix astragali* | Ethanol | MAE | [127] |
| Total flavonoids | Sea-buckthorn | None | MAE | [128] |
| Isoflavones | Soya bean | 50% methanol in water | MAE | [129] |
| Flavones | Cortex fraxini | Polyethylene glycol | MAE | [130] |
| Total flavonoids | Apple | Ethanol | MAE | [131] |
| Total flavonoids | Rosemary leaves | Methanol | MAE | [132] |
| Total flavonoids | Guava (*Psidium guajava*) | Ionic liquids (1-Butyl-3-methylimidazolium chloride + 1-butyl-3-methylimidazolium tetrafluoroborate) | MAE | [133] |
| Flavonols | *Phyllanthus emblica* | Water | MAE | [134] |
| Isoflavones | *Radix Puerariae thomsonii* | 70% methanol in water | MAE | [135] |
| Isoflavones | *Pueraria lobata* Ohwi | 70% ethanol in water | MA-UAE | [136] |
| Isoflavones | *Dalbergia odorifera* T. Chen | Aqueous two-phase system (ATP) | MA-ATPE | [137] |
| Isoflavones | Pigeon pea | 30% of water in 1,6-hexanediol/choline chloride (7:1) | DES-MAE | [138] |
| Antocyanins and flavonol | Apples | Methanol | PLE | [139] |
| Anthocyanins | Jabuticaba (*Plinia cauliflora*) | Ethanol | PLE | [123] |
| Flavones | Parsley | Ethanol:water, 50:50, and/or acetone:water, 50:50 | PLE | [140] |
| Anthocyanines | Cabbage (red) | Water/ethanol/formic acid (94:5:1) | PLE | [141] |
| Flavonones | Citrus | Water | PLE | [142] |
| Flavonols | Tea leaves; grape seeds | Methanol | PLE | [143] |
| Anthocyanins | Black carrots | Water acidified with lactic acid | PLE | [144] |
| Anthocyanins | Apples | Acidified methanol | PLE | [145] |
| Flavonols and flavan-3 ols | *Rheum palmatum* L. | 80% methanol in water | PLE | [146] |
| Isoflavones | *Trifolium* L. | 75% methanol in water | PLE | [147] |
| Isoflavones | Licorice | Ethanol | PLE | [54] |

**Table 4.** *Cont.*

| Flavonoids | Source | Solvent | Extraction Technique | Reference |
|---|---|---|---|---|
| Isoflavones | Soybean | 80% ethanol in water | PLE | [148] |
| Anthocyanins | Cabbage (red) | Water | PHWE | [141] |
| Flavonones | Citrus | Water | PHWE | [142] |
| Anthocyanins | Red grapes | Water | PHWE | [145] |
| Flavonole and flavonones | Oregano | Water | PHWE | [149] |
| Flavonols | Sea-buckthorn | Water | PHWE | [150] |
| Total flavonoids | Wine grape seeds | $CO_2$ | SFE | [151] |
| Total flavonoids | Maritime pine | $CO_2$ and 10% ethanol | SFE | [152] |
| Antocyanins and Flavonol | Grape bagasse | $CO_2$ and 10% ethanol | SFE | [153] |
| Flavonols and Flavones | *Mentha spicata* L. leaves | $CO_2$ | SFE | [103] |
| Isoflavones | Soya bean | $CO_2$ and 80% methanol | SFE | [154] |
| Isoflavones | Soya bean | $CO_2$ and 10% acetonitrile | SFE | [155] |
| Flavan-3-ols | Grape seeds | $CO_2$ and ethanol | SFE | [156] |
| Anthocyanins | Grape berries | $CO_2$ and ethanol | SFE | [157] |
| Anthocyanins | Grapes | $CO_2$ and ethanol | SFE | [158] |
| Anthocyanins and non-anthocyanin flavonoids | Grape pomace | Water | EAE | [159] |
| Isoflavones | Soya bean; chickpeas; *Trifolium subterraneum* L. leaves; *Lupinus albus* seeds; *Mentha piperita* leaves | Water | EAE | [86] |
| Isoflavones | *Trifolium pratense* L. | Dichloromethane/methanol (25:75) | MSPD | [160] |
| Flavonols | *Carthamus Tinctorius* L. | Methanol/water (1:3) | MSPD | [161] |
| Flavones and flavanones | *Murraya panaculata* (L.) Jack. | Methanol | MSPD | [162] |
| Isoflavones | *Dalbergia odorifera* T. | 66% ethanol | NPCE | [163] |
| Flavones | Pigeon pea | 70% ethanol | NPCE | [164] |
| Flavones | *Radix Scutellariae* | 75% ethanol | NPCE | [165] |
| Isoflavones | *Radix Astragali* | 70% ethanol | NPC-EAE | [166] |
| Flavonols | Flos *Sophorae immaturus* | 72% ethanol | NPC-UAE | [167] |
| Flavonols and anthocyanin | *Arabidopsis* plants | Buffer | In situ extraction with $TiO_2$ nanoparticles | [168] |
| Flavonones and flavanonols | *Sophora flavescens* | Solid reagent (basic salt) | MPET | [74] |

### 4.2.1. Ultrasound-Assisted Extraction (UAE)

UAE, also called sonication, uses ultrasonic wave energy during the extraction. Ultrasound produces cavitation, which accelerates the dissolution and diffusion of the cell ingredients: The ultrasound wave is propagated in the molecules of the medium and cavitation bubbles are formed at sufficiently high power. The disintegration of these bubbles generates energy, and the jets of solvent towards herbal particles extract the target compounds from them more efficiently [169]. In addition, the heat transfer improves the extraction efficiency. The other advantages of UAE include low solvent and energy consumption and reduction of extraction temperature and time, which make UAE a simple and relatively low-cost technology [170]. The most broadly used extraction apparatus is the ultrasonic bath. Because of the shorter processing time, this technique is relevant to extraction of thermally unstable compounds. This makes UAE an alternative to maceration and Soxhlet extraction. UAE has been employed in extraction of thermolabile compounds, such as anthocyanin from flower parts, in order to reduce extraction time and avoid high-temperature exposure [112]. But high ultrasound waves result in free radical formation and unacceptable changes of extracted components [13]. Therefore, all UAE parameters must be carefully optimized in order to avoid thermal degradation of phenolic compounds, such as flavonoids. The technique parameters are: Temperature, extraction time, polarity and amount of solvent, amount and type of sample, ultrasound frequency and intensity, and number of pulses. Lu et al. [113] optimized the operating conditions of UAE for total flavonoids from *Cryptotaenia japonica* Hassk using the Box–Behnken design. Their results suggested UAE as a promising technique for this extraction.

Different solvents and mixtures can be used to perform sonication, but most widely applied was ethanol (Table 4). Many reports demonstrated the application of UAE in the extraction of isoflavones. Isoflavones were extracted from *Pueraria lobata* (Wild.) Ohwi stem using 95% and 50% aqueous solutions of n-butanol and ethanol [114]. The authors established UAE as more efficient for the total yield of isoflavone extracts compared to Soxhlet extraction. The optimal extraction conditions were found to be: 50% ethanol (as extractant), 650 W ultrasonic power at 298.15 K with agitation rate at 300 rpm. Sun et al. [115] replaced the solvent by 40% ethanol for extracting puerarin from this plant material. The usage of an ethanol–water mixture for UAE also led to the successful quantification of puerarin, daidzin, daidzein, and genistein in the roots of *Pueraria lobata* and *P. thomsonii* [116]. Another study compared UAE to the classical techniques in the extraction of isoflavones from *Iris tectorum* [117]. The authors confirmed that UAE (in optimal condition: 70% methanol in water, 45 °C, 150 W, and 45 min) produced the highest extraction yields for the target isoflavones. Song et al. [118] isolated eleven major isoflavonoids from the xylem and bark of Radix Astragali by UAE using methanol. Rostagno et al. [170] compared UAE to conventional maceration in extraction of isoflavones from soya beans. The yield of the target compounds was found to be higher in the UAE extracts. The optimal UAE parameters were 333.15 K, 200 W sonication power, 20 min single run at 24 kHz frequency. Three different aqueous mixtures with ethanol, methanol, and acetonitrile (in the concentration interval from 30% to 70% *v/v*), and two temperature levels (10 and 60 °C) were tested. UAE was determined as a rapid and reliable technique. In this study, the ultrasonic bath was substituted by a probe horn and it yielded similar results. Pananun et al. conducted a very interesting experiment [126]. The researchers used high-power ultrasonication to extract isoflavones from defatted soybean with aqueous, acidified acetonitrile. The chosen parameters were 20 kHz ultrasound power, and changing amplitudes (18–54 μm) for 1 and 3 min. This process has not been studied in detail, but the extraction efficiency has been increasingly compared to conventional techniques.

Higher extraction efficiency compared to the simpler technique can be accomplished by a combination of various energy sources. A study conducted by Hu et al. introduced a new method for the rapid extraction of isoflavones from *Pueraria lobata* Ohwi [136]. They combined microwave and ultrasound energy sources. Isolation of isoflavones from plant material by ultrasound, combined microwave- and vacuum-drying technologies, proved to be a good alternative: The time required was 20-fold shorter than by reflux-extraction. The microwave-vacuum method additionally reduced the

processing time 10-fold. Moreover, this combined extraction technique showed no negative impact on the isoflavonoid content.

In conclusion, compared to the maceration and Soxhlet extraction, UAE has better merits such as time- and solvent-saving, requires lower temperature levels, and gives larger extract yields. However, the selectivity of this technique is comparable with that of the conventional extraction methods. The extract obtained contains a mixture of phenolic compounds and must be purified before the next step of identification or practical use [116,118], especially when fruits were subjected to extraction [85]. High-power sonication of soya beans reduced the total phenolic content, but the concentration of the major isoflavones (genistein, daidzein, and glycitein) increased 10-fold [126], so UAE can be a promising selective extracting technique with the proper extractant.

### 4.2.2. Microwave-Assisted Extraction (MAE)

MAE is an extraction technique that uses microwave power to promote the pouring of analytes from the sample matrix into the solvent. Microwave radiation works on the dipoles of polar and polarizable materials and causes heating near the surface of the materials. The heat is transferred by conduction. MAE can be considered as a selective method in the case of polar molecules and solvents with high dielectric constant [11]. Compared to conventional extraction methods, MAE offers a combination of advantages: The possibility to use fewer toxic solvents; reduced extraction time, solvent amount, energy, and processing costs; and increased recovery. Nevertheless, this technique is limited to small molecules stable under microwave heating conditions. Further added cycles of MAE led to lower yield of phenolics including flavanones, as a result of their oxidation [171]. Anthocyanins may not be suitable for MAE because of their potential thermal instability [171]. Table 4 lists the developed methods for MAE of flavonoids from several plant matrices. Laghari et al. [93] also investigated the extraction, identification, and antioxidative properties of flavonoids from *Cassia angustifolia* leaves and flowers and compared the efficiency of MAE and UAE to conventional extraction methods. The results obtained showed that the microwave extraction method is the better option for the total flavonoids extraction (Table 4). There is another example for MAE as a better extraction option, but not as a selective option: MAE conditions (solvent concentration, extraction time, and microwave power) for flavonoids and phenols extraction from Chinese quince (*Chaenomeles sinensis*) were optimized using designed experiments to maximize recovery by the method [172]. Different solvents can be used to extract flavonoids by this technique (Table 4): Water [134]; alcohol [127,130–132]; alcohol/water mixture [129,135]; or ionic liquids [133].

The extracts obtained by MAE are rich in some flavonoid fractions because of the plant material. For example, Rostagno et al. [129] obtained an extract from soya bean rich in isoflavones by MAE using methanol/water (1:1) as the solvent. However, MAE is not highly selective technique. Scientific literature reports studies on the extraction of flavonoids by MAE: Saponins [173], phenols [121,132,133,171], and coumarins [130]. Increased efficiency is associated with an increase in the amount of the flavonoid fraction and of the selectivity, respectively. Liu et al. [135] modified the MAE method as a first step for HPLC-fluorescence determination of isoflavones from *P. thomsonii* roots. The optimized extraction step was based on a 30-min sample soaking in 70% methanol, and a subsequent microwave exposure to 600 W for 11 min.

Two ecofriendly and cost-efficient variants of MAE were developed: Genistein and biochanin A were isolated from *Dalbergia odorifera* T. Chen leaves by microwave-assisted aqueous two-phase extraction (MA-ATPE) [137]. Ethanol, $K_2HPO_4/(NH_4)SO_4$/citrate, and deionized water built an aqueous two-phase system. First, a salt was dissolved in the water, then ethanol was added, and the solution was mixed until two phases were formed. The building of an aqueous two-phase system was attributed to the reciprocal exclusion of ions and ethanol on one hand, and their high affinity for the water on other hand. From the investigated salts, $K_2HPO_4$ was picked up, and it could be reused. Compared to conventional MAE and the under-reflux-extraction, the amount of target compounds in the extracts was higher by applying the new procedure. Another combined green technique for isoflavone extraction

from the roots of pigeon pea was investigated, which was named deep eutectic solvent-based MAE (DES-MAE) [138]. The optimal conditions for DES-MAE were designed by the single factor and Box–Behnken tests: 30% water in 1,6-hexanediol/choline chloride (7:1, n/n), 80 °C, microwave power of 600 W, and 11-min run cycle. The advantage of DES-MAE compared to the other extraction methods was the higher extraction efficiency. MA-ATPE and DES-MAE showed an enormous potential as environmentally friendly methods for the rapid and efficient extraction of isoflavonoids from plant samples.

### 4.2.3. Pressurized Liquid Extraction (PLE)

PLE has also been described as accelerated solvent extraction by different research groups. PLE applies high pressure, whereby the solvents remain liquid above their boiling point. The obtained effects are high solubility and high diffusion rate of hydrophobic compounds in the solvent, and high penetration of the solvent into the matrix [15]. Compared to other methods described above in this review, PLE requires less extraction time and solvent consumption and has better repeatability. Moreover, this technique can be automated which allows extraction in a shorter time with a minimum amount of solvent. For the first time, the Dionex Corporation introduced it in 1995 [174]. Contrary to the claim of some researchers that high temperature and pressure destroy the thermolabile compounds, Gizir et al. [144] successfully applied PLE to obtain an anthocyanin-rich extract from black carrots using acidified water as the solvent. The authors shared the opinion that the degradation rate of anthocyanins is time dependent. They fixed temperatures up to 100 °C, under 50 bar pressure, and a short processing time of a few minutes to overcome the possibility of molecular destruction. For this technique the choice of solvents is critical. The extraction must be as selective as possible, but solvents must be harmless, of lower toxicity, and easy to remove [13]. Solvent mixtures (e.g., methanol/ethanol–water) were found to be more effective and environment-friendly than pure organic solvents (Table 4). A successful optimal extraction of anthocyanins from lyophilized red grape skins was carried out by PLE using acidified water as a solvent. The acidified methanol was picked up from six solvents due to the reached maximum recovery rates of anthocyanins and total phenolics in the extracts tested. The optimal PLE conditions were 333.15 K and 10.1 MPa. This led to the decreasing of the solvent consumption and a total extraction time of 3 min [145]. After optimization, Santos et al. reported improvements in the recovery of anthocyanins (by 13%) and other phenolic compounds (by 8%) compared to the method of percolation under low pressure [123]. The optimized PLE process had the following parameters: 5 MPa, 553 K, 9 min, and static mode. Similarly, in another study [140], PLE was used for optimized extraction of flavones from parsley flakes. The researcher team used two-binary solvent systems: 50:50 *v/v* (ethanol/water and acetone/water). The flush volume significantly reduced the extractant consumption and the waste amounts obtained were lower. High recoveries (~94%) of flavonols and falvan-3-ols from *Rheum palmatun* were observed using 80% aqueous methanol by PLE [146]. Because of that, the method was suggested for quality control application. Chang et al. [148] investigated the influence of the ethanol/water ratio and the effects of the physiochemical parameters, including solvent flow rate and feed loading, on the efficiency of PLE of isoflavones from soybean flakes. Using 80%, aqueous ethanol achieved 95% recovery of the extraction method. Another study described the use of PLE in the micropreparative isolation of isoflavones from plants of the genus *Trifolium* L. [147]. Different extraction parameters (e.g., type of solvents, temperature, and the number of extraction cycles) were studied. The optimal extraction efficiency was achieved using methanol–water (75:25, *v/v*) at 125 °C. In this study, PLE showed significant advantages over conventional solvent extraction and UAE: Lower cost, high yield of isoflavone aglycones, relatively high precision and accuracy. Many other research teams have reported successful applications of PLE in flavonoid extraction from different plant matrices (Table 4).

Pressurized hot water extraction (PHWE) is a variation of PLE. This technique is also known as pressurized low-polarity water extraction and supercritical water extraction: The operating characteristics of the pressurized water used are temperature range above the boiling point and

below the critical point of water, which are 273 K at 0.1 MPa and 647 K at 22.1 MPa, respectively [13]. Under subcritical conditions, the intermolecular hydrogen bonds of water break down, and its dielectric constant decreases. At a temperature level of 250 °C, this constant is reduced to 27 [175], which is similar to the dielectric constant of ethanol (*varepsilon* = 24) and methanol (*varepsilon* = 33). The replacement of ethanol or methanol with water significantly reduces costs and makes this technique more environmentally friendly. The selectivity of this method relies on the solubility of the flavonoids which depends on the temperature conditions. The aqueous solubility of a variety of hydrophobic compounds increases at temperatures above 373.15 K and the water remains incompressible with further pressure rising under the operating conditions. High temperature has some unacceptable consequences, including degradation of thermolabile phenolic compounds and the necessity of further purification. In summary, the careful optimization of extraction temperature is important for PHWE. For example, PHWE resulted in enhanced extraction of flavonols from Seabuckthorn at 150 °C [150] and anthocyanins from dried red grape skin at 100–110 °C [145]. Rodriguez–Meizoso et al., [149] in an extensive research, concluded that more polar flavonoids, such as flavonols, flavones, flavonones, and anthocyanins, can be selectively extracted at a temperature of 100–120 °C. Above 120 °C and at long-term exposure, the flavonoids are destroyed. PHWE has industrial applications: Ko et al. [176] conducted a pilot-scale commercialization of the process. The effects of extraction parameters (temperature, pressure, time, material type, and sample/solvent ratio) on PHWE of flavonoids from dried satsuma mandarin peel were studied. The research team found the following optimal conditions: 130 °C, 15-min extraction time, and solute/solvent ratio of 1/34. The yield of flavonoids obtained under laboratory and pilot conditions was similar: 117.8 and 113.4 mg/g, respectively. The flavonoids recovery of PHWE in the pilot plant was 96.3%, which demonstrated the potential of this method for industrial application.

PHWE is not required for quantitative extraction of isoflavones [177]. Even at elevated temperatures and moderate pressure, water is not as effective as methanol, ethanol, and their mixtures with water for the isolation of isoflavones [87]. Moreover, the degradation of glucoside and malonyl forms was detected at higher temperature levels during PHWE [178,179].

### 4.2.4. Supercritical Fluid Extraction (SFE)

Supercritical fluid can be defined as a fluid above its specific critical temperature and pressure. Under these conditions, it exists in a phase, which has the properties of both liquids and gases at its critical point. Critical pressure is the minimum pressure required to convert a gas into a liquid state at its own critical temperature. The critical temperature of a gas is the temperature at which it does not become liquid under the application of extra pressure [13]. Temperature and pressure are the factors that determine the bringing of a substance into its critical region. The supercritical fluid behaves more like a gas, but has the solvating properties of a liquid. For example, $CO_2$ has this characterisic at above 31.1 °C and 7380 kPa. Supercritical-$CO_2$ (SC-$CO_2$) extraction has several very important advantages: Good solvating of non-polar analytes, low costs, and low toxicity of $CO_2$, and obtaining of a concentrated dry matter ($CO_2$ can be easily evaporated). To achieve greater solubility for polar compounds, a small amount of ethanol and methanol (defined as modifiers) were added. The huge disadvantage of this extracting technique was the use of an expensive installation.

Different bioactive flavonoid compounds, including catechin, epicatechin, rutin, myricetin, luteolin, apigenin, and naringenin, were obtained from spearmint (*Mentha spicata* L.) leaves using conventional Soxhlet extraction and SC-$CO_2$ extraction at different extraction schemes and parameters [103]. The effect of different parameters such as temperature (40, 50, and 60 °C), pressure (100, 200, and 300 bar), and dynamic extraction time (30, 60, and 90 min) on the SC-$CO_2$ extraction of spearmint flavonoids was investigated. The extracts obtained by Soxhlet and and the optimized SC-$CO_2$ extraction were further analyzed by HPLC to determine their chemical profile. Comparable results were obtained with the optimal SC-$CO_2$ extraction condition (60 °C, 200 bar, 60 min) and 70% ethanol as a solvent. Soxhlet extraction showed a higher crude extract yield (257.67 mg/g) compared to the SC-$CO_2$ extraction (60.57 mg/g). SC-$CO_2$ extract contained more main flavonoid compounds (seven

bioactive flavonoids) in a higher concentration compared to 70% ethanol Soxhlet extract (five bioactive flavonoids). Therefore, SC-CO$_2$ extraction is considered as an alternative selective process regarding the bioactive flavonoid compounds of spearmint leaves.

The efficiency of SC-CO$_2$ regarding polar analytes may be increased by adding a modifier [180]. The solvating properties of the fluid could be adjusted by optimization of pressure and/or temperature, which led to a relatively high selectivity [177]. The most important factor for the efficiency of the extraction method was the high solubility of the target substances in the supercritical extractant. Therefore, many parameters that affect the solubility and the resulting yield had to be taken into account. The optimal operating conditions for the phenolic compound extraction from grape marc and elderberry were investigated [156]. The researchers obtained extracts with high anthocyanin content. They used ethanol, ethyl-acetate, and acetone in different ratios with water as the modifier in a single-step mode. Temperatures of 20, 40, and 60 °C were applied. The influence of medium pH on the yield and degradation of anthocyanins was investigated, and two-step extractions, combining SFE and conventional batch extractions, were applied, too. Mixtures of organic solvent and water at 60 °C were shown to be the most effecive conventional solvents in single-step extractions. Pre-treatment of the natural material with SC-CO$_2$ (with or without ethanol as co-solvent) improved the extraction of polyphenols from the grape marc. This method provided a promising environmentally friendly alternative to the pre-treatment of the plant materials. Acidification of the medium resulted in a higher anthocyanin content in the extracts. However, the degradation of the anthocyanins during storage was higher, which led to the loss of the intensive color. Ghafoor et al. [158] used SC-CO$_2$ modified by 7% ethanol for the extraction of total phenolic compounds. The anthocyanin fraction was dominant. In a similar study of Murga et al. [156], a grape seed extract rich in phenols, including (+) catechin and (−) epicatechin, was obtained using the SC-CO$_2$ technique with ethanol as a modifier. In a study of Rostagno et al., an enhanced extraction efficiency of soy isoflavones (daidzein, genistein, daidzin and genistin) was obtained using higher modifier concentrations (80% methanol in water) [154]. The authors explained these results with an increase of CO$_2$ polarity and nearly constant density. The highest isoflavone yields were achieved by increasing the pressure (higher density of SC-CO$_2$) and CO$_2$ flow rates (higher mass of SC-CO$_2$). Raising the temperature (from 40 to 70 °C) can enhance the solubility of other matrix components and decrease the yield of isoflavones. Araújo et al. [155] found the optimal conditions of SFE for soybean isoflavones. An earlier step of the plant material preparation was thermohydration at 50 °C and pH 5.0. The researchers investigated the impact of various parameters (temperature, pressure, and added co-solvent) and compared the results obtained to the traditional solid–liquid extraction with 80% methanol in water. The SC-CO$_2$ results showed the highest yield of the target isoflavones at the following parameters: 60 °C, 380 bar, extraction time 15 min, and 10% acetonitrile. To extract isoflavonoid glycosides, a modifier addition was not enough. HPLC analysis quantification of daidzein and genistein showed that through solid–liquid extraction a higher quantity of daidzein and genistein was obtained than the amount acquired by SFE. For this reason, SFE cannot be recommended for quantitative isoflavone extraction.

4.2.5. Enzyme-Assisted Extraction (EAE)

EAE is an enzymatic pre-treatment which is carried out by the addition of specific hydrolyzing enzymes (e.g., cellulase, $\alpha$-amylase, and pectinase) during the extraction step [77]. By using enzymes, the cell wall is broken and the structural polysaccharides and lipid bodies are hydrolyzed under moderate conditions, which is reflected in high yields. This approach is considered a novel and effective technique for the extraction of a large group of secondary plant metabolites with antioxidant properties, including flavonoids [14]. In this regard, several parameters must be taken into account in order for the extraction process to be effective: Reaction temperature, extraction time, pH of the system, enzyme concentration, and the particle size of the substrate. Maier et al. [159] used a mixture of pectinolytic and cellulolytic enzymes (2:1 ratio, 2-h treatment, at 40 °C, and pH 4.0) to extract bioactive compounds including anthocyanins and non-anthocyanin flavonoidsfrom grape pomace. The yield obtained

was higher compared to sulfite-assisted extraction. The methodology led to isolation of phenol-rich fractions. Wallace and Burong [86] developed a selective EAE-procedure for isoflavone extraction from soya beans, chickpeas, *Trifolium subterraneum* L. leaves, *Lupinus albus* seeds, and *Mentha piperita* leaves using the hydrolyzing enzymes of the plants treated: The first step was the suspension of the ground material in water, followed by the warming (up to 62 °C at the treatment of *Trifolium subterraneum* L leaves) and waiting for the hydrolysis to finish (from 10 min to a few hours or overnight). Afte that, the pH value had to be adjusted up to 9,6–12 to obtain a soluble isoflavone fraction, and, finally, the precipitation of isoflavones from the filtrate was carried out by pH adjustment up to 3.5–5.6 and the precipitated isoflavone fraction was filtrated. The precipitate could be dissolved in ethanol with subsequent modification by the addition of acetone, whereby all dissolved sugars, saponins, and proteins were expected to be more or less precipitated.

The EAE is suitable for the extraction of various bioactive substances from plant matrices, but the fraction obtained after filtration is rich in water-soluble small molecule compounds, including flavonoids. After this procedure, it is possible to isolate the flavonoids step by step through changing the pH and adding different solvents [86]. However, this makes EAE a non-selective method regarding flavonoid extraction from plant materials.

### 4.2.6. Matrix Solid-Phase Dispersion (MSPD)

A popular alternative to the solid–liquid extraction method is MSPD. By means of this technique, the target compounds are extracted, fractionated, and prepared [177]. The process involves simultaneous homogenization, extraction, and purification, during which most of the problems associated with the classical methods are eliminated. Visnevschi–Necrasov et al. [160] developed a MSPD method for extraction and determination of 12 isoflavones in *Trifolium pratense* L.: Dried leaf samples were blended with hydrophobic C18-resin, placed in small columns, and isoflavones were extracted with a mixture of dichloromethane and methanol. Contrary to the standard extraction techniques, MSPD needs no specific labor equipment, less solvent, and it is fast, inexpensive, and more environmentally friendly. The selectivity depends on the eluent choice. Hong et al. [161] developed a MSPD method for the simultaneous extraction and isolation of kaempferol and other polyphenols such as hydroxysafflor yellow A from *Carthamus tinctorius* L., using silica gel as a dispersing sorbent (sorbent/sample ratio 3:1), and methanol:water (1:3, *v:v*). Graphene-encapsulated silica was used as an adsorbent in MSPD of polymethoxylated flavonoids from dried *Murraya paniculata* (L.) Jack leaves with recovery above 92.61% [162]. Compared to another five sorbents (graphene, silica gel, C18-resins, diatomaceous earth, and neutral alumina), graphene-encapsulated silica demonstrated better extraction efficiency regarding the target analytes.

### 4.2.7. Negative Pressure Cavitation Extraction (NPCE)

NPCE is a new type of cavitation, suitable for the isolation of thermo-unstable plant byproducts [177]. The main point of the process is a continuous introduction of nitrogen stream into the extraction system. Because of the negative pressure, $N_2$-bubbles arise in the liquid–solid system and a highly unstable gas–liquid–solid phase is built. These processes promote the turbulence, collision, and mass transfer between the extractant and the plant matrix, which leads to an effective extraction of the target analytes. During the process, a low temperature is kept, which makes it effective, simple, low-cost, and eco-friendly. NPCE shows potential for industrial application [181].

There is another advantage of this method: At room temperature processing, thermosensitive compounds such as isoflavonoids can be quantitatively extracted [163,166].

Dong et al. [165] isolated flavones from Radix Scutellariae by NPCE at the following optimal technological parameters: 40 mL/g solid, 75% ethanol in water, 60 min extraction time, and −0.07 MPa vacuum degree. The researchers concluded that NPCE has a good extraction efficiency compared to the other conventional extraction methods. The efficiency of NPCE increased significantly in combination with ultra-sonication. Wang et al. [167] developed a UAE method for the extraction of flavonols from

Flos *Sophorae immaturus* combined with negative pressure cavitation (NPC-UAE). The authors reported that the NPC-UAE extraction yield for six compounds was 1.27–1.62 and 1.17–1.40 folds higher than the output of NPCE and UAE, respectively. The optimal extraction conditions were: 72% ethanol, 16-min treatment time, liquid-to-solid ratio 25:1 mL/g, ultrasonic intensity 0.347 W/cm$^2$, negative pressure −0.07 MPa, and temperature 60 °C.

### 4.3. Other

Zhang et al. [74] developed a new method of flavonoid extraction based on mechanochemical-extraction technology. Physico-chemical transformations, generated by mechanical force, were used to obtain a solid fraction rich in flavonones and flavanonols: Pulverized *S. flavescens* roots and solid reagents (basic salts) were co-ground in a stainless-steel ball planetary mill for 10 min. After extraction with water for 20 min, the solution pH was adjusted to 4–5 by citric acid. Next followed a condensation and precipitant analysis. The highest yield of 35.17 mg/g was achieved by grinding the roots with $Na_2CO_3$ (15%) at 440 rpm/min for 17.0 min and using 25 mL solvent (water) per gram of solid material. The authors concluded that the extracts obtained by MPET, including kurarinol, kushenol I/N, and kurarinone determined by HPLC-MS/MS, showed good selective extraction.

Kurepa et al. [168] reported a preparation of nanoparticles, which deliver in vivo bioactive agents to their target cells: The anatase $TiO_2$ nanoparticles smaller than 20 nm enter plant cells, readily conjugate flavonoids rich in enediol and catechol groups in situ, and exit plant cells as nanoparticle-flavonoid conjugates. Nano-harvesting has a huge advantage: It eliminates the use of organic solvents. This approach opens new options for the use of the nanomaterials for simultaneous isolation and testing of bioactive properties of plant-synthesized compounds. The described technique for flavonol and anthocyanins extraction has even more advantages: Selectivity, no sample preparation, and preparation of materials for direct use. Futhermore, flavonoid nanoparticles can be easily prepared from limited amounts of material from different plant organs. Flavonoid/nanoparticle complexes obtained under sterile conditions in physiologically compatible buffers may be used for the treatment of mammalian cells immediately after the plant cell-based isolation step. Furthermore, flavonoid nanoconjugates isolated from various sources may also serve as a selection platform for screens aimed at the identification of flavonoid-binding biomolecules.

## 5. Conclusions

Plants are an inexhaustible source of flavonoids. The most important step in the plant raw material processing is the extraction and isolation of target compounds. A lot of research teams compared classical and modern methods for the extraction of flavonoids from plant materials and discussed their pluses and minuses. The focus of our study was the selectivity of the reviewed methods. Moreso, selectivity is equal to less time consuming, cost-effective, and environmentally friendly. The sample preparation and the extraction steps are equally important in the development of a selective extraction method of flavonoids.

Usually flavonoids and polyphenols are co-extracted by the application of classical techniques. When plants are a rich source of flavonoids, the extracts obtained are rich in flavonoids as well. The efficiency of extraction is higher by grinding the plant material and using alcohols as extractants. The extraction of thermo unstable flavonoids is dependent on the temperature and the treatment time. By applying green extraction techniques, such as MAE, UAE, and MSPD, the resulting yields are large, but this concerns both flavonoids and polyphenols. These techniques can be promising methods for selective flavonoids extraction with the proper extractant. PLE was more effective than MAE and UAE by using a methanol–water or ethanol–water medium. More polar flavonoids, such as flavonols, flavones, flavonones and anthocyanins, could be selectively extracted at temperature 100–120 °C by PHWE. Furthermore, PHWE had industrial applications, but was not required for the extraction of isoflavones. The interest in SC-$CO_2$ extraction was due to its high efficiency for non-polar analytes and the low cost and low toxicity of $CO_2$. A relatively high selectivity of anthocyanins, flavonols,

and flavonones extraction could be achieved by pressure and temperature modifying. However, SFE was not recommended for quantitative isoflavone extraction. EAE was a pre-extracting step rather than selective extraction procedure, because it was always followed by liquid–liquid extraction or another enrichment step. At room temperature processing, thermosensitive compounds such as isoflavonoids could be extracted quantitatively by NPCE. The application of NPCE in combination with UAE for the extraction of flavonols and flavones had a better extraction efficiency compared to the other conventional extraction methods, NPCE and UAE. Two separate methods developed—using anatase $TiO_2$ nanoparticles, and a method based on mechanochemical-extraction technology—are promising techniques for the selective extraction of flavonoids from plant matrices.

In conclusion, there is no universal extraction method and each extraction procedure is unique to certain plants. The design of sample preparation and extraction methods must be consistent with the study objectives, samples, and target compounds. For the extraction technique to be selective, it must incorporate an optimal solvent or mixture of solvents and a suitable technique. Last but not least, its optimization is important for a variety of applications, e.g., preparation of extracts for pharmaceutical usage, with a minimum level of impurities. In addition, when the method selected as the most appropriate one needs to be standardized, it must achieve an acceptable degree of repeatability and reproducibility.

**Author Contributions:** M.T., V.A., Z.Y., D.I., and T.D. contributed equally to the manuscript compilation and revisions. All authors have read and agreed to the published version of the manuscript.

**Funding:** This work was supported by the Bulgarian Ministry of Education and Science under the National Research Programme "Healthy Foods for a Strong Bio-Economy and Quality of Life" approved by DCM # 577/17.08.2018.

**Conflicts of Interest:** The authors declare no conflict of interest.

## Abbreviations

The following abbreviations are used in this manuscript:

| | |
|---|---|
| UAE | Ultra sonic-assisted extraction |
| MAE | Microwave-assisted extraction |
| MA-ATPE | Microwave-assisted aqueous two-phase extraction |
| DES-MAE | Deep eutectic solvent-based microwave-assisted extraction |
| PLE | Pressurized liquid extraction |
| PHWE | Pressurized hot water extraction |
| SFE | Supercritical fluid extraction |
| SC-CO$_2$ | Supercritical-CO$_2$ |
| EAE | Enzyme-assisted extraction |
| NPCE | Negative pressure cavitation extraction |
| LDL | Low-density lipoprotein |
| SPE | Solid phase extraction |
| HPLC | High performance liquid extraction |

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
