# Peer review of "Selectivity of Current Extraction Techniques for Flavonoids from Plant Materials"

_processes, doi:10.3390/pr8101222_

Round 1

Reviewer 1 Report

The work concerns a very wide group of natural compounds from the flavonoid group. Characterizes the extraction methods used to isolate them. As shown, there are many extraction methods. The paper is a nice overview of extraction techniques. But I have a few comments:
- I think it would be very useful to provide an overview comparison of the conditions (solvent, time) and yield  (preferably in tabular form on a selected example) of classical extraction techniques as well as greener variants such as the UAE, MAE, SFE extraction with supercritical fluids and other if possible.
- The article requires editorial revision
- Figure 2 could be made with slightly better quality

Author Response

The authors are very grateful for the useful comments and suggestions that greatly contribute to the quality of this work!

I think it would be very useful to provide an overview comparison of the conditions (solvent, time) and yield (preferably in tabular form on a selected example) of classical extraction techniques as well as greener variants such as the UAE, MAE, SFE extraction with supercritical fluids and other if possible.”

  • Yes, it would be very useful to provide an overview comparison of the conditions of the both groups extraction methods (classical and greens), but this should be a topic of another review work, and I think such was already published by other authors (e.g. Processes 2020, 8(4), 434; https://doi.org/10.3390/pr8040434).

 “Figure 2 could be made with slightly better quality.

  • Figure 2 was replaced by another file format.

Reviewer 2 Report

The paper deals with the selectivity of different extraction techniques for flavonoids from plant materials.

The topic of this research is interesting, and the paper is quite well-organized. However, some changes and additions are necessary to make some parts more understandable.

The quality of most of the Figures and Tables should be improved.

P41-43 Please reformulate the sentence to better explain and add a reference.

P48-49 Please add a proper reference.

P50-58 Please insert some references.

P64 Please remove the quotation marks and move the bibliographic reference at the end of this sentence.

P87-88 Please change font and size of the text according to the journal instructions.

Table 1 must be reedited because is not very understandable. Please separate the table and make one table relating to food sources and another one to medicinal plants.

P112-114 Please add a reference.

P116-118 Please add a reference.

Figure 2: Please change the font of Figure 2 according to the journal instructions. Please correct “ Food supplements” in the first box.

Table 2: Formatting of Table 2 is not great. Please improve it to make reading easier.

P246 Please better explain the cavitation process by ultrasounds. Some important references in the field of UAE should be added. In example:

[1] M. Vinatoru, T.J. Mason, I. Calinescu. Ultrasonically assisted extraction (UAE) and microwave assisted extraction (MAE) of functional compounds from plant materials, TrAC Trends in Analytical Chemistry,Volume 97, 2017, Pages 159-178, https://doi.org/10.1016/j.trac.2017.09.002.

[2] Farid Chemat, Natacha Rombaut, Anne-Gaëlle Sicaire, Alice Meullemiestre, Anne-Sylvie Fabiano-Tixier, Maryline Abert-Vian. Ultrasound assisted extraction of food and natural products. Mechanisms, techniques, combinations, protocols and applications. A review, Ultrasonics Sonochemistry, Volume 34, 2017, Pages 540-560, https://doi.org/10.1016/j.ultsonch.2016.06.035.

P265 Please replace the two points with the point.

P246-308 Paragraph 4.2.1 need a reorganization. Difference between extraction by ultrasonic bath and probe must be considered and better explained. Moreover, the sentence “Ultrasonic baths can be replaced by a probe horn as it yields similar results” is not correct. Generally, direct UAE by probe (or sonotrode) is more efficient and leads to higher extractive yields. Please reformulate this paragraph highlighting the differences between direct UAE (probe) and indirect UAE (bath).

P320-321 “Anthocyanins may not be suitable for MAE….” Why? Please add some references to justify this sentence. 

Author Response

The authors are very grateful for the useful comments and suggestions that greatly contribute to the quality of this work!

“P41-43 Please reformulate the sentence to better explain and add a reference.”

  • The sentence was reformulated.

"P48-49 Please add a proper reference”.

  • Proper references were added.

“P64 Please remove the quotation marks and move the bibliographic reference at the end of this sentence.”

  • The quotation marks were removed, and the reference was moved to the end of this sentence

“P50-58 Please insert some references."

  • A proper reference was inserted.

“P87-88 Please change font and size of the text according to the journal instructions”

  • The font and size of the text were changed.

“Table 1 must be reedited because is not very understandable. Please separate the table and make one table relating to food sources and another one to medicinal plants.”

  • Table 1 was divided into two separated tables.

“P112-114 Please add a reference.”

  • A proper reference was added.

“P116-118 Please add a reference.”

  • A proper reference was added.

“Figure 2: Please change the font of Figure 2 according to the journal instructions. Please correct “Food supplements” in the first box.”

  • Figure 2: The font is changed, and “Food supplements” is corrected.

“Table 2: Formatting of Table 2 is not great. Please improve it to make reading easier.”

  • Because of the splitting of Table 1, Table 2 became Table 3. So, Table 3 was revised.

“P246 Please better explain the cavitation process by ultrasounds. Some important references in the field of UAE should be added.”

  • A short explanation of the cavitation process was added that referred to the suggested reference (https://doi.org/10.1016/j.trac.2017.09.002).

“P265 Please replace the two points with the point.”

  • The two points are replaced with one point.

“P246-308 Paragraph 4.2.1 need a reorganization. Difference between extraction by ultrasonic bath and probe must be considered and better explained. Moreover, the sentence “Ultrasonic baths can be replaced by a probe horn as it yields similar results” is not correct. Generally, direct UAE by probe (or sonotrode) is more efficient and leads to higher extractive yields. Please reformulate this paragraph highlighting the differences between direct UAE (probe) and indirect UAE (bath).”

  • Yes, generally, direct UAE by sonotrode is more efficient. However, it depends on the biological material and the extracted target compounds. In Paragraph 4.2.1. was reported a UAE method for extraction of isoflavones from soya beans., and in that case it “yields similar results”.

“P320-321 “Anthocyanins may not be suitable for MAE….” Why? Please add some references to justify this sentence.”

  • The statement was explained and a proper reference was added.

Reviewer 3 Report

The article presented by the authors is well organized and can be a useful guide to the extraction of flanovoids.

However, some of the techniques are not well explained and the English should be checked for typos and grammar inaccuracies.

More specific comments are presented in the following points:

Author's names: Please check the author's names. It seems that there are some typos.

Line 38: The authors should present the definition of LDL in the text the first they use the abbreviation of this lipoprotein.

Lines 38 and 39: These sentences are somehow confusing. When flavonoids prevent the oxidation of LDL molecules while protecting them, this will avoid atherosclerosis? It is the oxidized LDL that is responsible for atherosclerosis?

Line 113: A reference is missing in this sentence for freeze-drying. The authors found no references for the use of a food-dryer or a dehydrator? A dehydrator has different characteristics from other drying methods and should be included.

Figure 2: The diagram presents a few typos: “Suppliments” change to supplements; “research and develop of flavonoid…” change to “research and development of flavonoid…”; “Ultra sonic” to “ultrasonic”;

Line 156: The authors should include a reference in this paragraph.

Line 161: The authors should include references in this paragraph.

Table 2: Instead of a column with a repetition of the extraction technique, the authors could add a line with the technique before each application list and eliminate this column. Also, in the last line of the table, the authors should start hexane with an upper case.

Table 3: Please see the comments for table 2. Also, change “Puffer” to “Buffer”.

Line 310: The authors should always use the same abbreviations for MAE. MAP can infer different conditions or types of equipment, and this is not the case. Processing implies several techniques and not just extraction.

Line 326: Please use the abbreviation for MAE instead of the definition.

Line 405: Change the punctuation (: to .).

Line 428: The definition of SWE should be presented previously in Lines 407 and 409 when the authors present the subcritical conditions. The authors should also explain the difference between the different subcritical methods: solvents; pressure, temperature, more clearly.

Lines 530 to 534: The authors should include references in this paragraph.

Line 541: Please change C18-rasin to C18-resin.

Line 550 to 552: Please add a reference.

Line 585 to 599: References are missing

Line 646: According to the text, the list of abbreviations is uncomplete. If the authors always explain the definition of these abbreviations in the text this table can be deleted.

Author Response

The authors are very grateful for the useful comments and suggestions that greatly contribute to the quality of this work!

“Author's names: Please check the author's names. It seems that there are some typos.”

  • Author's names were checked, there are not typos.

“Line 38: The authors should present the definition of LDL in the text the first they use the abbreviation of this lipoprotein.”

  • LDL definition was added.

“Lines 38 and 39: These sentences are somehow confusing. When flavonoids prevent the oxidation of LDL molecules while protecting them, this will avoid atherosclerosis? It is the oxidized LDL that is responsible for atherosclerosis?”

  • Oxidized LDL is known to associate with the development of atherosclerosis, and it is therefore widely studied as a potential risk factor of cardiovascular diseases. (Stocker, Roland; Keaney, John F. (2004). "Role of Oxidative Modifications in Atherosclerosis". Physiological Reviews. 84 (4): 1381–1478. doi:10.1152/physrev.00047.2003. PMID 15383655). LDL has the important task of transporting the “bad” cholesterol and the neutral lipids. So, when LDL are destroyed/oxidized, they can be deposited in the walls of the arteries and become part of atherosclerotic plaques.

“Line 113: A reference is missing in this sentence for freeze-drying.

  • The proper reference was added.

“Figure 2: The diagram presents a few typos: “Suppliments” change to supplements; “research and develop of flavonoid…” change to “research and development of flavonoid…”; “Ultra sonic” to “ultrasonic””

  • The corrections were done, please see Figure 2.

“Line 156: The authors should include a reference in this paragraph.”

  • The proper reference was added.

Line 161: The authors should include references in this paragraph.

  • The proper reference was added.

Table 2: Instead of a column with a repetition of the extraction technique, the authors could add a line with the technique before each application list and eliminate this column. Also, in the last line of the table, the authors should start hexane with an upper case.”

  • Because of the splitting of Table 1, Table 2 became Table 3. So, Table 3 was revised.

Table 3: Please see the comments for table 2. Also, change “Puffer” to “Buffer”.

  • Because of the splitting of Table 1, Table 3 became Table 4: “Puffer” was corrected. It would not be very appropriate to remove the column “Extraction technique” because of the presence of some special cases of technique combination.

“Line 310: The authors should always use the same abbreviations for MAE. MAP can infer different conditions or types of equipment, and this is not the case. Processing implies several techniques and not just extraction.”

  • The abbreviation was corrected.

“Line 326: Please use the abbreviation for MAE instead of the definition.”

  • The abbreviation was used instead of the definition.

“Line 405: Change the punctuation (: to .).”

  • The punctuation was changed.

“Line 428: The definition of SWE should be presented previously in Lines 407 and 409 when the authors present the subcritical conditions.”

  • This was a typo and was removed.

“Lines 530 to 534: The authors should include references in this paragraph.”

  • A proper reference was added.

“Line 541: Please change C18-rasin to C18-resin.”

  • C18-rasin was changed to C18-resin.

“Line 550 to 552: Please add a reference.”

  • A proper reference was added.

“Line 585 to 599: References are missing”

  • A proper reference was added.

“Line 646: According to the text, the list of abbreviations is incomplete. If the authors always explain the definition of these abbreviations in the text this table can be deleted.”

  • The Abbreviation list was completed. In the main text, the definition was replaced by the abbreviation.

Reviewer 4 Report

Submitted manuscript is well written and easy to read. However, the scope of the review was to large, and resulted in narration sounding similar to a chapter in monograph. Moreover, surprisingly little discussion was focused on selectivity that was emphasized in the manuscript title.  Provided literature is sufficient and includes 179 references, but less than half of them were published within recent 10 years and lest than 15% covers last 5 years. The biggest concern relates to novelty of the manuscript, recently similar review article (slightly less abundant in extraction methods) was also published in the  Processes 2020, 8(4), 434; https://doi.org/10.3390/pr8040434 .

Author Response

The authors are very grateful for the useful comments and suggestions that greatly contribute to the quality of this work!

“Submitted manuscript is well written and easy to read. However, the scope of the review was to large, and resulted in narration sounding similar to a chapter in monograph. Moreover, surprisingly little discussion was focused on selectivity that was emphasized in the manuscript title. 

  • In our review we have used published data about extraction methods of flavonoids from food and medicinal plants, and made conclusions about their selectivity. What is the selectivity important for? “The potential role of selective extraction methods” has a significant importance for the subsequent steps of the extract production. The selectivity of this first extraction step contributes to the effectivity, profitability and simplicity of the whole process of extract production.

Provided literature is sufficient and includes 179 references, but less than half of them were published within recent 10 years and lest than 15% covers last 5 years.”

  • Flavonoids are molecules long known and their extraction also. So, we didn’t want to ignore the authors who are the pioneers in the field, especially regarding the biological activity of flavonoids and the classical extraction techniques, which are still up to date.

“The biggest concern relates to novelty of the manuscript, recently similar review article (slightly less abundant in extraction methods) was also published in the  Processes 2020, 8(4), 434; https://doi.org/10.3390/pr8040434.”

  • This just proves the actuality of the subject. In the reference list can be seen also other related reviews. The work cited above is focused on the biotechnological methodologies for flavonoid extraction from agro-industrial residues.

Reviewer 5 Report

The manuscript studies Selectivity of current extraction techniques for flavonoids
from plant materials. It is well written and well organised. It can b e recommended for publication after some minor changes:

  1. line 4 check the name if it is correct (saMilena)
  2. line 60 refernces could be cited as 11-16
  3. table 1 delete fulstop after [58]
  4. table 1 ref 74 not in red color.
  5. line 138 replace word algorithm as not correct 
  6. line 178 and throughout text when giving 0.1% HCl with methanol, define whether it is volume ratio
  7. Please check all text for Italics when needed in species.

Author Response

The authors are very grateful for the useful comments and suggestions that greatly contribute to the quality of this work!

line 4 check the name if it is correct (saMilena)

  • The name was checked. There was no typo.

line 60 refernces could be cited as 11-16

  • The references were cited as 11-16.

table 1 delete fulstop after [58] & “table 1 ref 74 not in red color.”

  • Due to the division of Table 1, the corrections were made in Table 2.

line 138 replace word algorithm is not correct

  • Figure 1 heading: “algorithm” was replaced by “scheme”

line 178 and throughout text when giving 0.1% HCl with methanol, define whether it is volume ratio

  • Yes, it is volume ratio, it was added.

Please check all text for Italics when needed in species.

  • The whole main text was checked for Italics when species are in consideration.

Round 2

Reviewer 3 Report

The authors answered all the comments and reviewers' questions. After the correction of some typos the article can be published in the present form.

Reviewer 4 Report

The manuscript has been substantially improved. Unfortunately, most of the flaws indicated in my previous review were not changed nor addressed properly in authors response. Nevertheless, the manuscript can be recommended for publication. Please correct some editorial mistakes that were made during revision of the manuscript i.e. line 118, 124, 275, 367, 407 and 414 (use °C as through the manuscript), 476, 629.